

# Doping-dependent competition between superconductivity and polycrystalline charge density waves

Sergio Caprara[1], Marco Grilli[1], José Lorenzana[1] and Brigitte Leridon[2*]

**1** ISC-CNR and Department of Physics, Sapienza University of Rome,
Piazzale Aldo Moro 2, I-00185, Rome, Italy
**2** LPEM, ESPCI Paris, CNRS, Université PSL, Sorbonne Universités,
10 rue Vauquelin, 75005 Paris, France

⋆ Brigitte.Leridon@espci.fr

## Abstract

From systematic analysis of the high pulsed magnetic field resistance data of $La_{2-x}Sr_xCuO_4$ thin films, we extract an experimental phase diagram for several doping values ranging from the very underdoped to the very overdoped regimes. Our analysis highlights a competition between charge density waves and superconductivity which is ubiquitous between $x = 0.08$ and $x = 0.19$ and produces the previously observed double step transition. When suppressed by a strong magnetic field, superconductivity is resilient for two specific doping ranges centered around respectively $x \approx 0.09$ and $x \approx 0.19$ and the characteristic temperature for the onset of the competing charge density wave phase is found to vanish above $x = 0.19$. At $x = 1/8$ the two phases are found to coexist exactly at zero magnetic field.

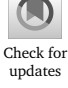
# 1 Introduction

The phase diagram of high-$T_c$ cuprate superconductors encompasses a rich variety of behaviors and competing phenomena. The antiferromagnetic Mott-insulating state that characterizes the undoped parent compound is rapidly disrupted upon doping, and the Néel temperature vanishes in the hole-doped compounds for doping $p \geq 0.02$, while the superconducting critical temperature $T_c$ rises and then decreases. The density of states at the Fermi energy is also partially suppressed in all cuprate families below a crossover pseudogap temperature $T^*$ [1], with $T^*(p)$ decreasing with increasing $p$ and reaching $T_c$ around optimal doping. Polarized neutron scattering experiments have highlighted a time-reversal and inversion symmetry breaking below $T^*$ in YBa$_2$Cu$_3$O$_{6+\delta}$ (YBCO) [2–6] that could be mirrored in the superconducting state symmetry [7]. This symmetry breaking was later reproduced in HgBa$_2$CuO$_{4+\delta}$ (HBCO) [8–10], La$_{2-x}$Sr$_x$CuO$_4$ (LSCO) [11] and Bi$_2$Sr$_2$CaCuO$_{8+\delta}$ (BSCCO) [12,13].

Under high magnetic field the topology of the Fermi surface is deeply affected in the underdoped region and magnetotransport studies [14–17] find a Fermi surface reconstruction in YBCO for doping between $p \approx 0.08$ and $p \gtrsim 0.16$. More recent work on the same compound also suggests that a sharp change in the number of carriers occurs around the end-point of $T^*$ at $p \approx 0.19$ [18][1] signaling a restructuring of the Fermi surface with a substantial reduction of its volume at low doping. In the same conditions, a static charge ordering (CO or static CDW) is observed by NMR [20] and by X-ray scattering [21] at $T$ lower than $T_c(p)$. Above $T_c$ resonant X-ray scattering [22–28] experiments have detected fluctuating charge density waves (CDW) and NMR [29] experiments points to static short-range CDW pinned by the existing disorder, even at low magnetic field.

This variety of behaviours naturally involves an interplay of physical mechanisms and a variety of theoretical proposals. First of all the pseudogapped phase below $T^*(p)$ might be due to a novel exotic metallic state driven by strong correlations in the proximity of the Mott insulating phase [2] or it might arise from some quantum critical point (QCP) underneath the superconducting dome associated to some ordered state: circulating currents [31,32], or charge order (CO) [33–38]. Some authors argue that magnetic fluctuations can play a role [39] despite the magnetic quantum critical point being located in the underdoped region outside the superconducting dome. The presence of Cooper pairs is another physical ingredient that should be considered to describe the properties of the pseudogapped state. Although the existence of *stable* preformed Cooper pairs below $T^*$ has been questioned [40,41], measurement of paraconductivity effects in the pseudogap state in LSCO [40, 42, 43] show that standard Cooper pair fluctuations are present in a large temperature range above $T_c$.

In this Article, we analyze resistance data under high magnetic field for LSCO thin films with various dopings, ranging from heavily underdoped to heavily overdoped ($0.045 \leq x \leq 0.27$). We use large magnetic fields to tip the balance between different phases across the entire doping range. The phase diagram is in excellent agreement with a scenario in which, at low doping, disorder induces filamentary superconductivity (SC) inside an otherwise charge-ordered phase [44]. Therefore, for the sake of concreteness we refer to the high field phase as CDW. One should keep in mind, however, that magnetotransport can not univocally identify the order parameter and other scenarios involving different forms of order might explain the data [30,45]. On the other hand, it remains a challenge for those scenarios to explain the peculiarities of the experimental phase diagram that we present below and which are well explained within a charge ordered scenario.

A minimum at $T_{MIN}$ in the temperature-dependence of the resistivity separating a negative slope at low $T$ from a positive slope at larger $T$ at $p < 0.17$ and high $H$ was evidenced long

---

[1]This is also in accord with an early scenario based on optical conductivity computations [19]

[2]For a review see, e. g., [30].

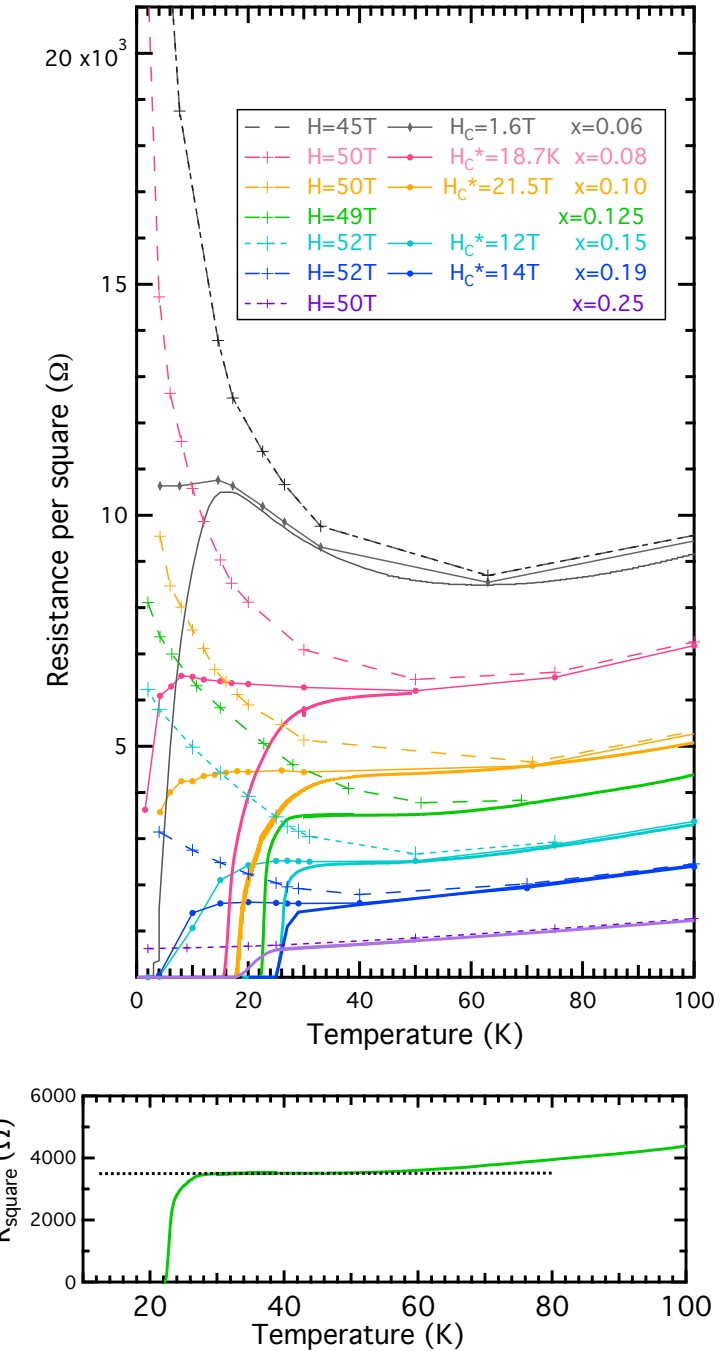

Figure 1: Top: Resistance per square for different Sr content $x$ = 0.06, 0.08, 0.1, 0.125, 0.15, 0.19, 0.25: at zero field (solid line), at the critical field $H_C^*$ for which there is a high-temperature plateau (thin solid line with dots), and at the maximal field, between 45 T and 52 T, depending on the sample (dashed line with crosses). For $x$ = 0.06, only $H_C$ is attainable (see text). Bottom: Resistance per square for the $x$ = 0.125 sample, evidencing a plateau already present at $H = 0$ between about 28 K and 55 K. The dashed line is a guide to the eye.

ago [46]. Here we attribute this minimum to the onset of polycrystalline CDW (producing the negative slope logarithmic contribution at low temperatures) and we extend this analysis at

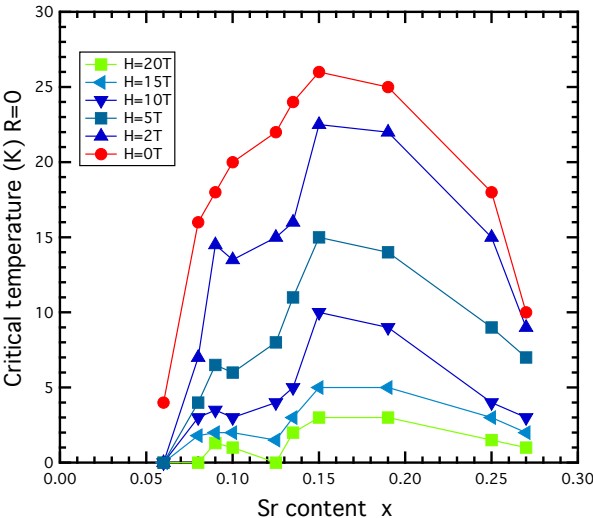

Figure 2: Superconducting critical temperature $T_c$ ($R = 0$) as function of the Sr content for different values of the magnetic field. Note the presence of two distinct domes at high magnetic field, centered approximately around $x = 0.09$ and $x = 0.19$, as previously observed in YBCO in Ref. [49]

intermediate fields, evidencing a much richer situation: besides this minimum, we investigate the occurrence of an inflection point $T_{INF}$ marking the onset of strong Cooper pair fluctuations eventually driving the system superconducting below a low-$T$ maximum $T_{MAX}$. Interestingly, at specific critical values of the magnetic field $H_c^*$ we also find plateaus in $R(T)$ that highlight a quantum critical behaviour separating the competing CDW and the superconducting phases. In particular we find a peculiar double-step superconducting transition, already observed in Refs. [44, 47, 48], and studied here systematically as a function of doping. The phase diagram in the $(H, T)$ plane for different dopings looks remarkably similar with the main difference that the transition lines move rigidly to higher or lower fields depending on doping. Thus at each doping a different observation window to a generic phase diagram is accessible.

## 2 Measurements

The sample preparation and characterization are described in Refs. [47, 50]. Pulsed high-field resistance measurements up to 45–52 T were performed from 1.5 or 4.2 K up to 300 K on several LSCO thin films with different Sr content $x$, spanning across the underdoped, optimally doped, and overdoped regions of the $T$ vs. $x$ phase diagram. The geometry of the samples and the details of the measurements are given in [44, 47].

The top panel of Fig. 1 displays the resistance $R$ vs. $T$. Each color corresponds to a different doping. For each value of $x$ we show three curves: the zero field resistance (thick solid line) the maximum field resistance (dashed line) and the resistance for and intermediate field $H_C^*$ (thin solid line). The latter is defined by the appearance of a plateau in the $R(T)$ curve, over a more or less extended temperature range, depending on doping. For $x = 1/8$ the plateau appears at $H_C^* = 0$ (as shown in the lower panel) and for higher dopings there is no plateau and only two curves are shown.

We interpret the plateau as due to quantum critical behavior between an insulating state and a superconducting state over an extended temperature range. For a conventional QCP, this

behavior would extend down to zero temperature and intersect the resistance axis at a finite value. We see in Fig. 1 that, on the contrary, at low temperature a different mechanism takes over and the system turns into the superconducting state so the QCP (called here QCP1) is in reality an "avoided QCP". Still, for each doping we can define the $H_C^*$ corresponding to QCP1 by extrapolating the finite temperature data. Alternatively, when plotted as a function of $H$, the $R(H)$ curves for different temperatures in the plateau range intersect in a point identifying the critical field $H_C^*$ and resistivity $R_c$ (see the inset of Fig.9a in Appendix B and [47]).

As a function of doping we basically encounter three different situations. For overdoped samples ($x \geq 0.2$), the $R(H)$ curves never cross, the resistances at any field are all increasing functions of $T$, so that no plateau appears in the $R(T)$ curves as already mentioned (see, e.g., Fig. 6 in the Appendix for $x = 0.25$). For intermediate dopings ($0.08 \leq x \leq 0.19$), $R(H)$ curves cross at $H_C^*$ (QCP1) for a wide range of temperatures (Fig.9a) signaling QCP1. A second QCP2 appears at $H_C$ (with $H_C > H_C^*$) where superconductivity disappears at low temperatures, as was found (for $x = 0.09$) in Refs. [47] and [48]. In this doping region, the $R(T)$ curves are monotonic and ever increasing in the absence of magnetic field, while a resistance minimum appears when the magnetic field exceeds a doping dependent value (see for example the top orange curve in the top panel of Fig. 1). For lower dopings ($x \leq 0.06$) only the low temperature plateau appears at $H_C$ (thin black line in Fig. 1) with an associated low temperature crossing point in $R(H)$ (left panel of Fig. 7 in the Appendix for $x = 0.06$). The resistance $R(T)$ has a minimum already at $H = 0$ (see right panel of Fig. 7 in the Appendix ). For $x \leq 0.05$, the samples are not superconducting (see Fig. 8 in the Appendix), R(T) has a minimum and there is no magnetoresistance.

A scaling analysis in the vicinity of QCP1 for four different samples revealed a set of critical indices $vz = 0.45 - 0.63 \pm 0.1$ (See Fig.9b in Appendix for $x = 0.09$ and [47].) This critical behavior eventually stops upon cooling, QCP1 is avoided and the system becomes superconducting. Superconductivity persists for fields larger than $H_c^*$ but with low critical temperatures. This situation of fragility to temperature combined with strong resiliency to fields was explained in Ref. [44] as due to a filamentary superconducting phase originating in competing real space and momentum space order in the presence of quenched disorder. In this theoretical scenario, QCP1 corresponds to the "clean" quantum critical point which would separate a zero temperature CO phase from a superconducting phase in the absence of disorder. Quenched disorder breaks the long-range CO into a polycrystalline phase. At the domain boundaries, CO is frustrated and superfluidity prevails so that the superfluid phase penetrates well into the CO stability domain in the $(H, T)$ plane and QCP1 is avoided [51]. When increasing the field, filamentary SC is eventually spoiled and a second low temperature plateau is reached at a field $H_C$, marking a second QCP (QCP2 in Fig. 5). A scaling analysis at QCP2 for $x = 0.09$ finds $vz = 1.0 \pm 0.1$ [47].

## 3 Phase diagram

In Fig. 2 we plot the critical temperature as function of the Sr content for different values of $H$. As already observed in YBa$_2$Cu$_3$O$_x$ (YBCO) [49] samples and also inferred in [52], what appears as a single dome at low field, splits up at higher fields into two distinct domes that are centered around respectively $x \approx 0.09$ and $x \approx 0.19$ (with a maximum at around $x \approx 0.15 - 0.17$). The separation between the domes occurs at the remarkable doping value $x = 1/8$. We discuss later the possible insights provided by this intriguing structuring of the phase diagram.

Next, we extract from the $R(T)$ curves the characteristic temperature scales, $T_{MIN}$, $T_{INF}$, and $T_{MAX}$ that we use as proxies for the onset of CO, of superconducting fluctuations and SC

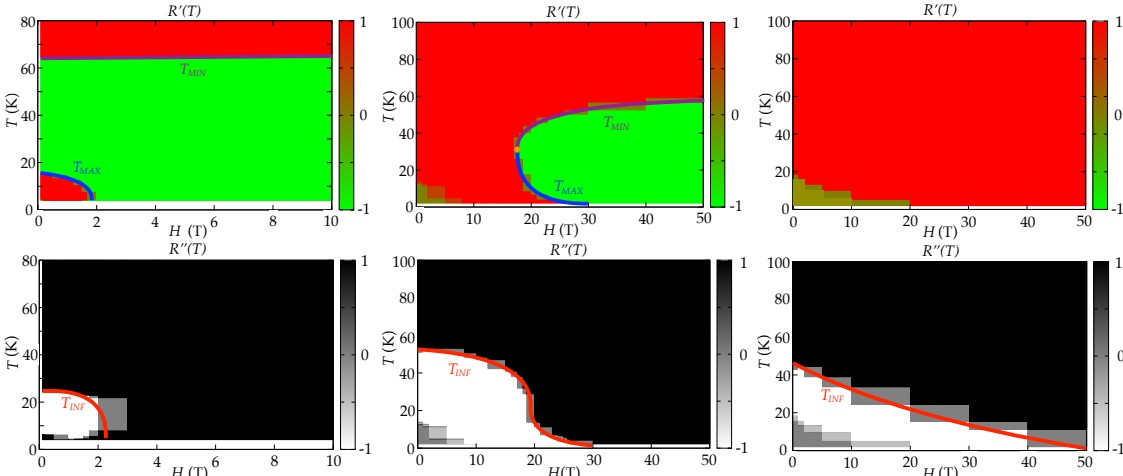

Figure 3: Top panels: Color maps of the sign of the sign of $R'(T)$ in the $(H,T)$ plane. As a guide to the eye, the $T_{MIN}$ lines are marked in purple, the $T_{MAX}$ lines are marked in blue and their common point (if any), is marked by an orange spot signaling the location of the plateau. Bottom panels: Color maps of the sign of $R''(T)$ in the $(H,T)$ plane. The $T_{INF}$ line is marked in red. Left panels: $x = 0.06$; $T_{INF}$ is always much lower than $T_{MIN}$ for this doping. Central panels: $x = 0.08$. Right panels: $x = 0.25$; note that at this doping $R'(T) \geq 0$ is always true.

as in Ref. [44]. To do so, we interpolate all the curves and calculate their first and second temperature derivatives. Except for long-range SC these are characteristic temperatures for crossover behavior and therefore their precise value for a specific doping and magnetic field is not very significant. What is significant is their dependence as a function of doping and magnetic field. Special care has been taken in defining those proxies in a way that does not depend on the accuracy of the measurement (i.e. we avoid defining onsets from a signal going below a detection threshold except for the zero resistance state in Fig. 2). This makes the dependence on doping or magnetic field of $T_{MIN}$, $T_{INF}$, and $T_{MAX}$ physically significant and allows to study systematically the form of the resulting phase diagram.

Some typical phase diagrams are displayed in Fig. 3 for dopings $x = 0.06$, $x = 0.08$ and $x = 0.25$, from left to right. For all dopings, the topmost graph is the color map of the sign of the first derivative $R'(T) = dR/dT$, in the $(H,T)$ plane (the green/red color marking a negative/positive derivative). The bottommost graphs are a color map of the sign of the second derivative $R''(T) = d^2R/dT^2$ (with black/white marking a positive/negative sign). The boundary line separating the two latter regions corresponds to an inflection point. The white color thus always corresponds to the superconducting region (either long range or fluctuating), which is the only phenomenon leading to a marked downward curvature [43]. These two maps allow to draw a phase diagram in the $(H,T)$ plane for each doping, which we illustrate in the three paradigmatic cases of Fig. 3.

For $x = 0.06$, $R(T)$ has a minimum at all $H$, so $R''(T) > 0$, except at low $H$ and $T$, where SC occurs. The line separating red and green regions in the top-left panel of Fig. 3 corresponds to the minimum and $T_{MIN}$ is independent of $H$. $T_{INF}$, which is given by the zero field value of the curve separating the black and white region in the bottom-left panel of Fig. 3 is always much lower than $T_{MIN}$. Of course, the transition to the superconducting state is anticipated by a maximum of $R(T)$ at $T_{MAX}$. For $x = 0.08$, the line separating the green and red in the top-center panel of Fig. 3 corresponds to a minimum ($T_{MIN}$) at high temperature and to a maximum ($T_{MAX}$) at low temperature, that coalesce into an inflection point with horizontal tangent (a plateau) at a finite $T$ and $H$. No minimum in the resistance is present at zero and

low magnetic field. Notice that the $x = 0.06$ case (left panel) resembles strikingly the high field part of the central panel. That is, the main effect of increasing doping is to shift the characteristic fields to higher values effectively moving the "window" in parameter space over which the phase diagram is probed. Sample $x = 0.25$ is qualitatively different as there is never a minimum in $R(T)$, at least up to the highest available fields.

We now analyze the behavior of the characteristic temperatures as a function of doping. In Fig. 4(a), the red dots represent $T_c(x)$ while $T_{INF}(x)$, marking the onset of superconducting fluctuations at zero temperature, is identified by the red dotted line. The green squares represent the position $T_{MIN}(x)$ for the resistance minimum at the highest measured $H$. $T_{MIN}(x)$ is found to drop abruptly to zero somewhere between $x = 0.19$ and $x = 0.25$.

We interpret $T_{MIN}(x)$ as the onset temperature for static short-range CDW [29] or polycristalline CO. At $T_{INF}$ SC fluctuations come into play and at lower temperature superconductivity prevails, resulting in an $R = 0$ state below $T_c$. At $x = 0.06$, this state is already filamentary, while at higher doping it becomes filamentary upon applying a magnetic field, above $H_c^*$. Fig. 4(b) pictures the critical field $H_C^*$ at QCP1 and the critical field $H_C$ at QCP2. For $x = 0.06$ we have a "weak" zero field filamentary SC which gets suppressed by a small field. At $x = 1/8$ at zero field the system is exactly on the verge between CO and SC, so at low temperature exactly on the verge between filamentary and bulk SC. Magnetic field then drives the system into the filamentary phase which persists up to high fields. This is consistent with the generally admitted idea that CO is very stable at $x = 1/8$. Away from $x = 1/8$ a finite field is needed to suppress SC and favor the CO state where, due to disorder, filamentary SC arises at the boundaries of polycristalline CO.

Based on the outcomes of our analysis, we can draw the phase diagram of Fig. 5, where a suitable control parameter can drive the occurrence of the CDW that competes with SC. At each doping level, one can explore only a limited portion of the whole phase diagram by tuning $H$. This is pictured by the green arrows in Fig.5, which indicate the physical accessible range of the phase diagram for a magnetic field $0\,T \leq H \leq 50\,T$. Note the particular case of $x = 1/8$, for which QCP1 exists already at $H = 0$.

## 4  Discussion

Based on the many evidences for CDW mentioned in the introduction and on theoretical arguments [34, 38, 53], in this work we identified the region of negative $R'(T)$ to a CDW phase. Now we elaborate on this, providing more specific arguments in favor of this identification. First of all a connection emerges between SC and the end-points of the static CDW region from the field-induced modifications of the SC dome reported in Fig. 2 (see also Ref. [49] for YBCO). In particular one might conceive two possible scenarios. The first possibility is that commensuration effects favour the CDW order at $x = 1/8 = 0.125$, giving rise to a dip in the $T_c(x)$ dome. Increasing $H$ weakens SC making this dip more and more pronounced until the SC dome is split in two. The second possibility is that increasing $H$ suppresses SC thus favouring the competing CDW state, but for the regions of the phase diagram around the end-points $x_1$ and $x_2$ of the CDW dome hidden under the SC dome. Here the strong charge fluctuations around the QCPs would favour pairing [54] rendering SC more resilient around $x_1 = 0.08 - 0.09$ and $x_2 = 0.15 - 0.16$. The occurrence of SC near QCP is a rather common phenomenon, e.g. in heavy-fermion systems [55].

The main evidence in favour of the identification of the phase with negative $R'(T)$ with a CDW phase arises from the occurrence of a low-temperature superconducting phase in the underdoped region, well inside the domain of stability for CDWs. This was recently explained in terms of disorder-promoted filamentary SC topologically protected at the domain bound-

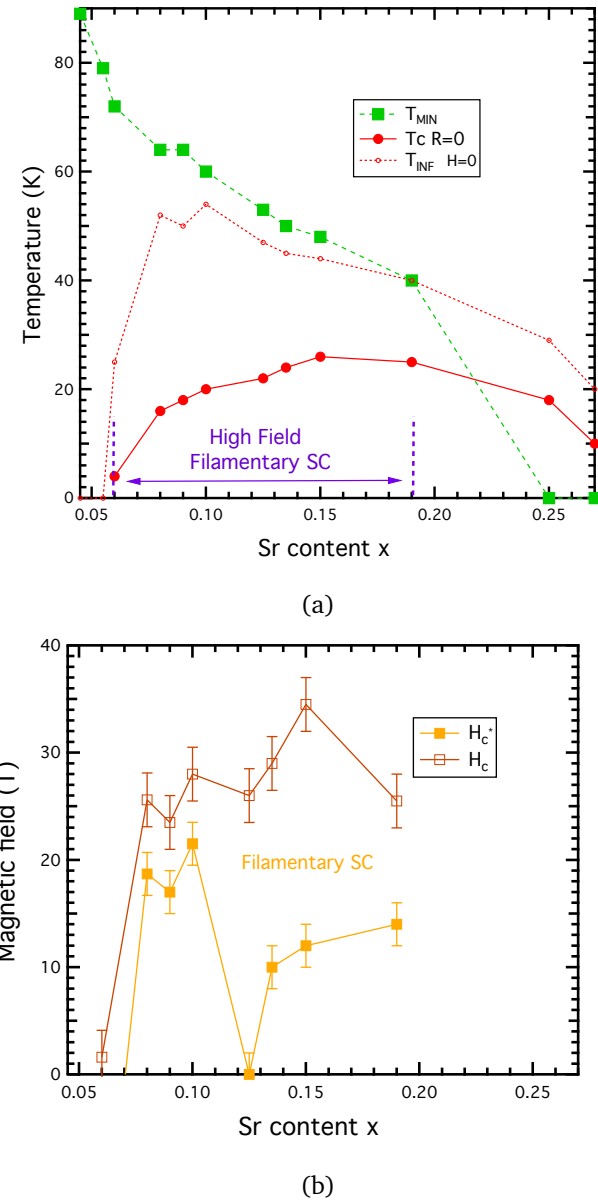

(a)

(b)

Figure 4: Characteristic temperatures as function of the Sr content. (a) Red dots: $T_c$ for $R = 0$; green squares: temperature of the minimum of $R(T)$ high field ($T_{MIN}$); Red dashed line, position of the inflection point at zero temperature $T_{INF}$. (b) Yellow filled squares : value of the magnetic field at the plateau $H_C^*$ corresponding to QCP1; brown open squares: value of the magnetic field at the low temperature plateau, corresponding to the actual superconductor/CDW critical field (QCP2). For clarity $H_C^*$ has been put arbitrarily to an imaginary value for x=0.06, in order to illustrate the fact that the observed SC is indeed only filamentary at this doping value. Note the abrupt change at $x \geq 0.19$ in $T_{MIN}$, where all filamentary SC disappears, and the singularity at $x = 1/8$ in $H_C^*$.

aries of CDW domains [44]. Experimental evidence for the coexistence of CDW domains and SC is also given in $HgBa_2CuO_{4+y}$ in Ref. [56]. The idea of filamentary superconductivity in cuprates was put forward long ago in Refs. [57,58].

Concerning the quantum critical regimes at $H_C^*$ and $H_C$ (orange dashed lines in Fig. 5) this

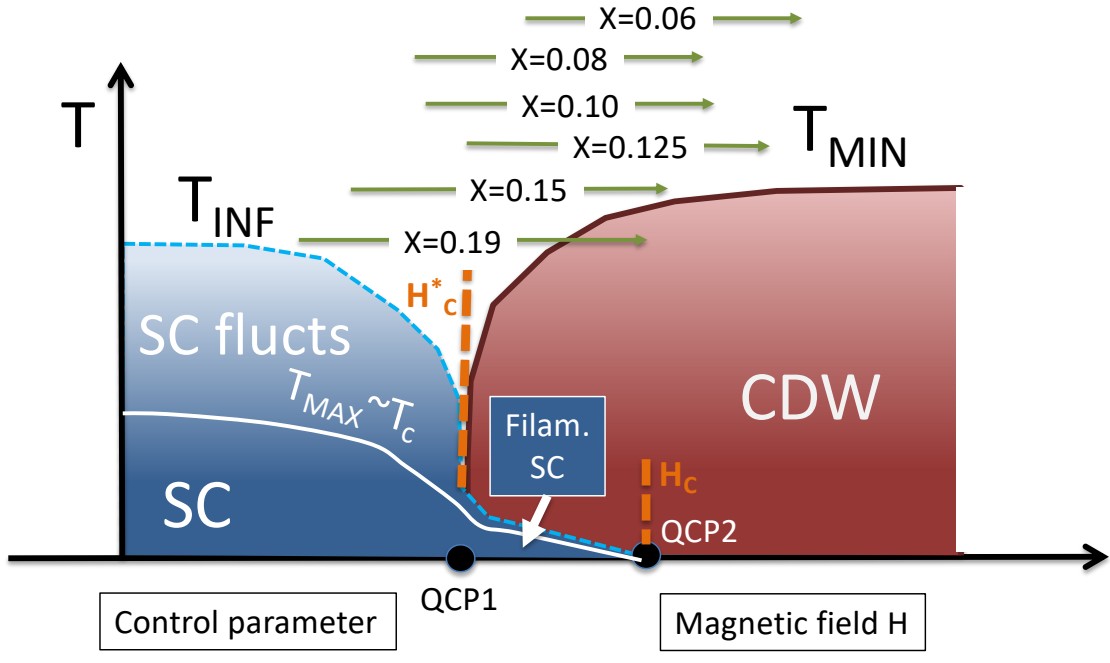

Figure 5: Proposed schematic phase diagram. On the horizontal axis the control parameter tunes the competition between SC (favored by low fields and/or high doping) and the CDW (favoured by high fields and low doping). The orange dashed line denotes the position of the plateau. As demonstrated in [44], disorder is responsible for the filamentary superconducting phase which penetrates deeply inside the domain of stability for CDW and renders the QCP1 invisible at low temperature. While entering further into the CDW a second QCP2 is reached. The green arrows schematically mark the range spanned by increasing the magnetic field at different dopings. The change in doping entails the observation window provided by the magnetic fields to slide over the phase diagram in a non-monotonic way. At $x = 1/8$, CDW and SC coexist exactly at $H = 0$.

two-step superconductor-insulator transition was already observed in a previous work [47] at $x = 0.09$. Later, this two-stage transition was attributed to properties of the vortex lattice, with the intermediate state for $H_C^* \leq H \leq H_C$ corresponding to a vortex glass for which $R = 0$ only at $T = 0$ [48]. We notice that a filamentary superconductor in the presence of a magnetic field will have vortices with several flux quanta located in random positions and therefore will not be an ordered lattice but will be indeed a glass. Notice, however that we do not find $T_c = 0$ but we find $R = 0$ over a finite temperature range for $H_C^* \leq H \leq H_C$ (see also Fig. 4(b) in Ref. [47]) i.e. $T_c$ is finite. We also notice that the critical field $H_C^*(x)$ at the plateau, which separates the CDW and SC stability domains is found to vary consistently with $T_c$ at low doping (with 1 T equivalent to 1 K), then drops abruptly to about zero at $x = 1/8$ and then increases again, as can be viewed in Fig4(b). This anomaly at $x = 1/8$, where commensurability effects are customarily believed to be relevant, also indicates that charge ordering interplays (and competes) in a relevant way with SC. The fact that the quantum critical region separating the competing charge order and the superconductivity (QCP1) is already present at zero field, exactly (and only) for $x = 1/8$, again, favours our identification of the competing phase with a CDW phase.

The above results were obtained for LSCO thin films. We briefly discuss here how they can

be relevant to other cuprates. First it is noteworthy that the dip in $T_c$ at $x = 1/8$ is a common feature of all hole-doped cuprates. And the splitting of the $T_c$ dome into two different domes observed at high field was previously observed in YBCO [49].The two-step transition with magnetic field has been observed to date only in LSCO in resistivity, but NMR and specific heat measurements have found a phase diagram in YBCO [59] which is remarkably similar to the transport phase diagram of Fig. 5 and the central panels of Fig. 3. The two-step transition in resistivity is expected to occur when CDW are present (which is the case in YBCO) and for sufficiently disordered systems. Indeed, it would be interesting to check in other cuprates whether the avoided quantum critical QCP1 in the resistivity data is also present at zero magnetic field for the "magical" value $x = 1/8$.

## 5  Conclusion

In summary, the complex behaviour of the resistivity throughout the doping-temperature-magnetic field phase diagram in LSCO thin films can be rationalised in terms of two simple ingredients: a) a competing, polycristalline CDW phase inducing the negative-slope behaviour in $R(T)$ at low $T$ and high fields and b) superconducting Cooper fluctuations that, upon decreasing the field, overcome the CDW state and induce SC. The effect of changing the doping level is to slide in a non-monotonic way the observation window provided by the magnetic field over the phase diagram for this two phases, as illustrated in Fig.5, so that for certain values of doping the critical region between the two phases is fully observable, while for some other dopings it is unaccessible. In addition, the doping decreases monotonically the strength of the CDW, as is illustrated by the green squares for $T_{MIN}$ in Fig.4(a). The balance between the two phases thus evolves in a non-monotonic way with doping, with a peculiarity at $x = 1/8$, for which they coexist exactly at $H = 0$. Due to disorder, the superconducting phase survives at low temperature in the domain of stability for CDW and acquires a filamentary character, which persists as long as a CDW phase is present, i.e. for $x \leq 0.19$.

## Acknowledgements

J.L. and S.C. thank all the colleagues of the ESPCI in Paris for their warm hospitality and for many useful discussion while this work was done. BL also gratefully thanks J. Vanacken and V.V. Moshchalkov for hospitality at KU Leuven. Part of the work was supported through the Chaire Joliot at ESPCI Paris. This work was also supported by EU through the COST action CA16218 NanocoHybri.

**Funding information**  S.C. and M.G. acknowledge financial support of the University of Rome Sapienza, under the Ateneo 2017 (prot. RM11715C642E8370) and Ateneo 2018 (prot. RM11816431DBA5AF) projects. J.L. acknowledges financial support from Italian MAECI through collaborative project SUPERTOP-PGR04879 and bilateral project AR17MO7 and from Italian MIUR through Project No. PRIN 2017Z8TS5B, and from Regione Lazio (L. R. 13/08) under project SIMAP. Part of the work was supported through the Chaire Joliot at ESPCI Paris. This work was also supported by EU through the COST action CA16218.

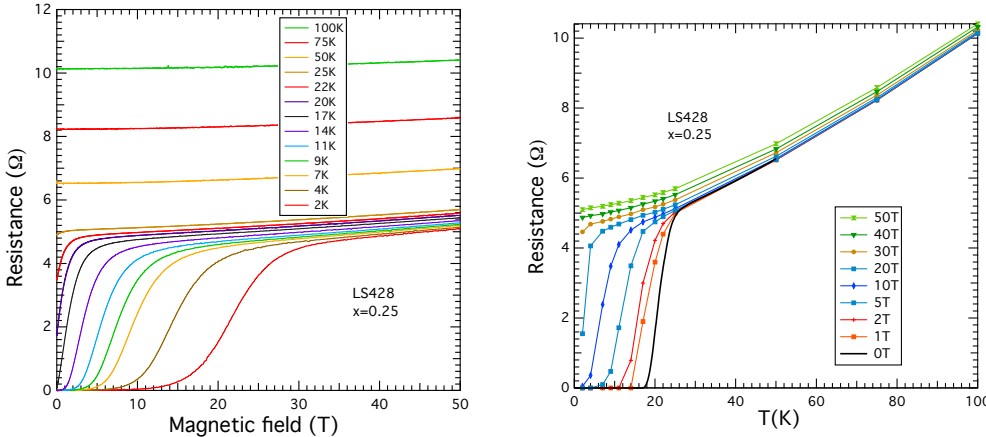

Figure 6: Overdoped sample $x = 0.25$. Left: Resistance as function of magnetic field. Right: resistance as function of temperature for different magnetic field values.

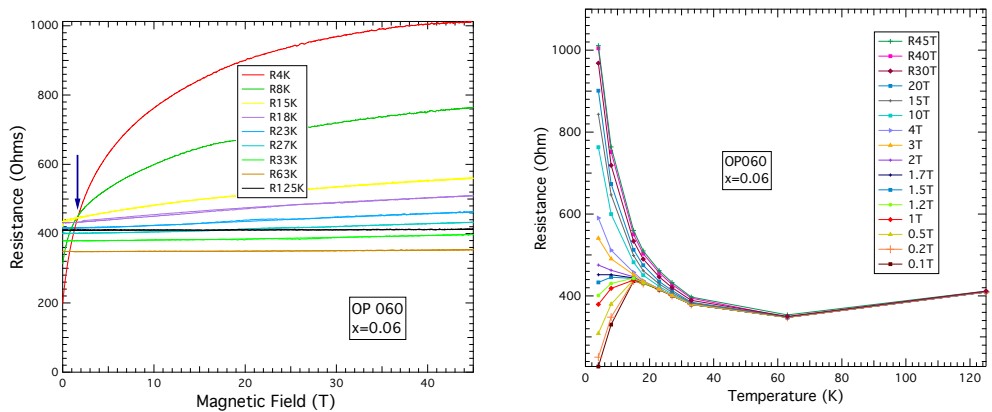

Figure 7: Underdoped sample $x = 0.06$. Left: Resistance versus magnetic field at different temperatures. Right: Resistance versus temperature at different magnetic fields. The crossing point is indicated in the left panel by an arrow and corresponds to the low temperature plateau at about 1.7 T in the right panel.

## A  Additional data about sample resistances

Resistance versus magnetic field and resistance versus temperature are displayed for an overdoped sample ($x = 0.25$) in Fig.6 and for an underdoped sample ($x = 0.06$) in Fig.7. Resistance for a strongly underdoped non-superconducting sample ($x = 0.045$) is plotted in Fig.8. In this case the magnetoresistance is negligible but the curve still presents a minimum. As may be seen in Fig. 1, this minimum is placed in continuity with all other dopings.

## B  Analysis of the quantum critical region in LSCO x=009

In Fig. 9a, the resistance versus temperature curves are plotted for x=0.09 for magnetic fields ranging from 0T to 47 T by steps of 1 T. The R(T) exhibits a plateau for $H_c^* = 17\,T$. This corresponds to a fixed point in the R(H) curves for $H = H_c^*$ (see inset of Fig. 9a) for $9\,K \leqslant T \leqslant 26\,K$. Scaling analysis was performed at the vicinity of the fixed point. Fig. 9b shows the scaling of the R(H) curves for sample LSCO0.009a , as $R/R_c^* = f(|H - H_c^*|T^{-1/\nu z})$ with $\nu z = 0.46 \pm 0.1$.

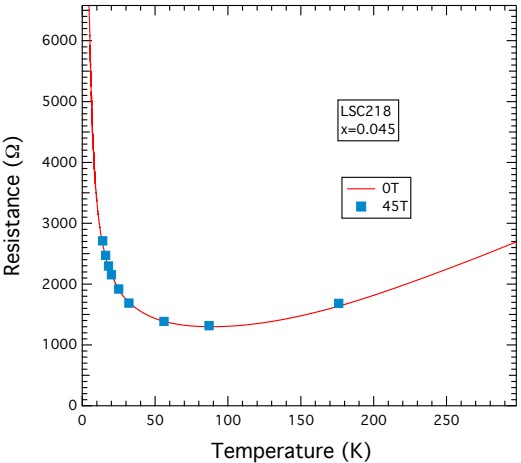

Figure 8: Underdoped non-superconducting sample $x = 0.045$. Resistance versus temperature at zero and 45 T magnetic field. There is no measurable magnetoresistance.

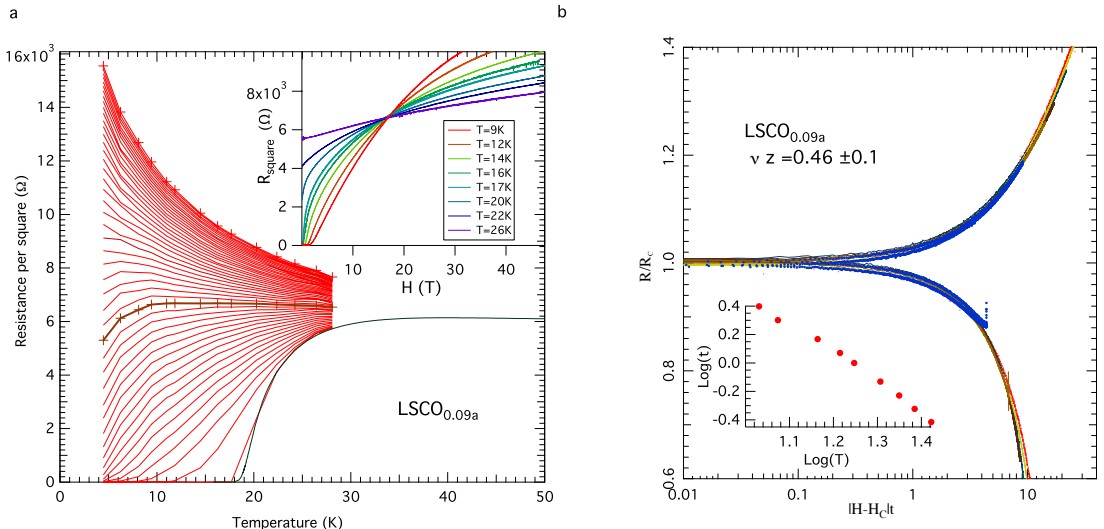

Figure 9: **a** R(T) data for different magnetic field values, ranging from 0 T to 47 T from bottom to top by steps of 1 T for sample $LSCO_{0.09a}/STO$ ($x = 0.09$). The brown line, showing a plateau from about 9 K to about 26 K, corresponds to $H_c^* \simeq 17\,T$. Inset: Corresponding R(H) data for temperatures between 9K (lower curve at low fields) and 26 K (upper curve at low fields). **b** Scaling of the R(H) curves for the same sample, for $R/R_C = f(|H - H_C|t)$, with $t = T^{-1/\nu z}$ and $\nu z = 0.46 \pm 0.1$. Inset: Log(t) versus Log(T).

The scaling exponent $\nu z$ was found to vary from $0.45 \pm 0.1$ to $0.63 \pm 0.1$ for the four different samples on which scaling was possible (x=0.08,0.09a, 0.09b and 0.1). According to the scenario in Ref. [44], this critical behaviour is associated to a symmetric Heisenberg model separating an easy-axis model representing the CDW phase from an easy-plane model representing the superconducting state. Various situations may however occur depending on the interplane coupling (which is different for SC and CDW). If this is is large enough to allow for a 3D Heisenberg model then a finite $T_c$ occurs. If, instead the interplane couplings are small $T_c$ is very low and a 2D quantum critical behaviour arises in the plateau temperature range. If

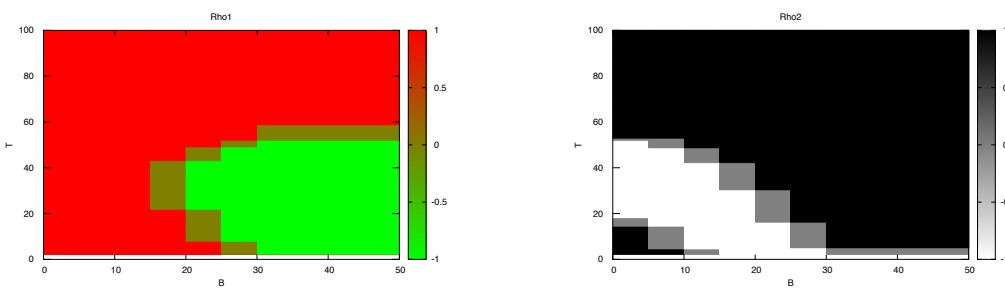

Figure 10: $x = 0.10$ a) Color map of the sign of the $R'(T)$ in the $(H, T)$ plane. b) Color map of the sign of the $R''(T)$ in the $(H, T)$ plane

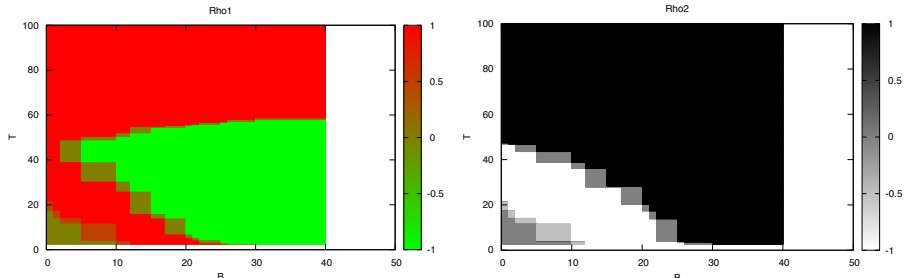

Figure 11: x=0.125 a) Color map of the sign of the $R'(T)$ in the $(H, T)$ plane. b) Color map of the sign of the $R''(T)$ in the $(H, T)$ plane

$T_c$ is finite, but somewhat lower than the temperatures of the plateau, one might even find an intermediate quantum 3D (i.e. 3+1 dimensional) behaviour. In the first two cases one would expect for the classical 3D or quantum (2+1) Heisenberg model the critical indices to be $z = 1$ and $\nu \simeq 0.7$. In the intermediate (3+1) case a mean-field set of indices is instead expected ($z = 1$ and $\nu \simeq 0.5$). Another possibility has also been indicated in Ref. [48], where the case of a clean (2+1) xy model was proposed. Concerning the critical behaviour around QCP2, the superconducting transition involves the phase locking between locally superconducting filaments. More theoretical and experimental work is needed to clarify this issue which goes beyond our present scope.

## C  Additional CDW/SC phase diagrams

Fig.10, 11, 12 and 13 picture the experimental phase diagrams for Sr content respectively $x = 0.10$, $x = 0.125$, $x = 0.15$ and $x = 0.19$. For all these doping values the phase diagram shows an excellent agreement with the phase diagram of Fig.5 and is therefore consistent with a coexistence of SC fluctuations and polycrystalline CDW, with a filamentary SC phase subsisting well inside the domain of stability for the CDW.

The phase diagram for $x = 0.125$ shows that the plateau (orange dashed line in Fig.5 of the main text) approaches zero magnetic field, while for dopings below and above, it is at finite magnetic field, pointing to the existence of a "ghost" or avoided quantum critical point at $x = 1/8$ and $H = 0$. Actually this diagram is obtained from interpolation from high-field data for which there is only a limited number of temperature points due to the very time

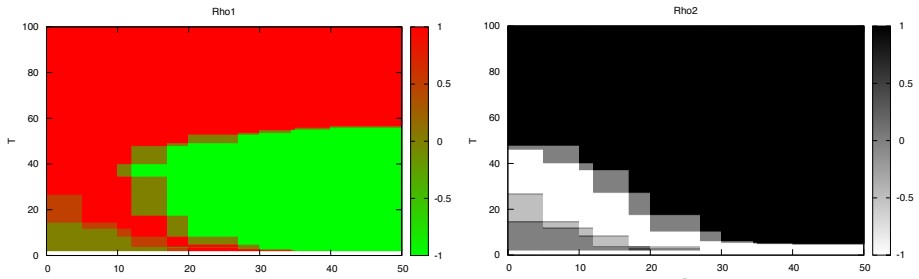

Figure 12: x=0.15 a) Color map of the sign of the $R'(T)$ in the $(H, T)$ plane. b) Color map of the sign of the $R''(T)$ in the $(H, T)$ plane

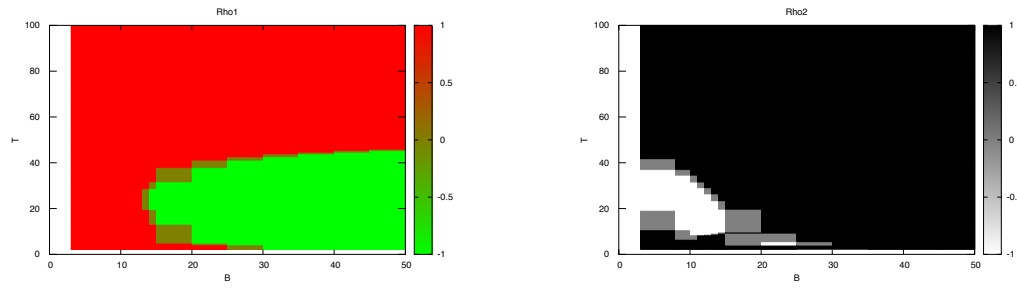

Figure 13: x=0.19 a) Color map of the sign of the $R'(T)$ in the $(H, T)$ plane. b) Color map of the sign of the $R''(T)$ in the $(H, T)$ plane

consuming aspect of these measurements. (One temperature point is about 4-5 hours.) So the plateau at zero field obtained after interpolation from this scarce data is not very well defined, although it is very clear from the resistance taken at zero field (with as many temperature points as necessary) that a plateau is already present at zero field in the $R(T)$ curve. (See Fig.1 bottom). This is not the case for any of the other dopings.

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
