# Peer review of "Doping-dependent competition between superconductivity and polycrystalline charge density waves"

_SciPost Physics, doi:SciPost Phys. 8, 003 (2020)_

## Round 2 · Referee Report · Anonymous · 2019-10-16

Strengths

A very good experiment ensured by the application of high magnetic fields.

Weaknesses

Missed the previous publications by J. C. Phillips.

Report

The article describes very informative experiments on LSCO carried out in high magnetic fields. These magnetic fields helped to reveal hidden charge order (possibly more or less conventional CDWs) in certain areas of the LSCO phase diagram. The results are impressive and I recommend the article for publication. Nevertheless, the article would become better if the authors follow the remarks given below.

Requested changes

1. When the experimental data are discussed in the Introduction, the specific materials should be indicated since the behavior of various superconductors is not universal and should not be universal.

2. The sentence “Of course spin fluctuations from the antiferromagnetic phase nearby can also play a role [29]” should be explained because antiferromagnetism dies out for dopings corresponding to the superconducting dome.

3. Measurements were carried out for LSCO. How to reconcile it directly with the data obtained for other cuprates? It would be interesting if authors briefly describe the distinctions and common features of the CDW-superconductivity relationship between the materials concerned.

4. The idea of the filamentary superconductivity in cuprates belongs to J. Phillips and his works must be cited. [see, e.g., J. C. Phillips, Phys. Rev. B, 43, 11415 (1991)].

---

## Round 3 · Author Response

Dear Editor,

We thank the referee for their careful reading of the manuscript as well as their fruitful comments and suggestions. We have taken all of them into account in the revised version.

With best regards,

Dr. Brigitte Leridon,
on behalf of all authors.

---

## Round 3 · List of Changes

We have added the information on the different materials in the introduction paragraph (and added subsequent references). We have reformulated the sentence about spin fluctuations,
in order to avoid ambiguity.
We have added references to Dr. Phillips' work [57,58].
We have also added a full paragraph discussing the differences between cuprate families in relation to our findings just before the conclusion.
The changes are printed in red in the revised version.

Resubmission 1908.03408v3 on 26 November 2019
Submission 1908.03408v2 on 7 October 2019

---

## Editorial Decision

published